behaviour/environmental science/acoustics

attract-and-kill, brown marmorated stink bug, Hemiptera, integrated pest management, Pentatomidae, substrate-borne signal

**Author for correspondence:**
Carol L. Bedoya
e-mail: clbedoya.contact@gmail.com

# Brown marmorated stink bug overwintering aggregations are not regulated through vibrational signals during autumn dispersal

Carol L. Bedoya[1], Eckehard G. Brockerhoff[1,2,3], Michael Hayes[4], Tracy C. Leskey[5], William R. Morrison III[6], Kevin B. Rice[7] and Ximena J. Nelson[1]

[1]School of Biological Sciences, University of Canterbury, Private Bag 4800, Christchurch, New Zealand
[2]Scion (New Zealand Forest Research Institute), Christchurch, New Zealand
[3]Swiss Federal Research Institute WSL, Birmensdorf, Switzerland
[4]Department of Electrical and Computer Engineering, University of Canterbury, Private Bag 4800, Christchurch, New Zealand
[5]USDA, Agricultural Research Service, Appalachian Fruit Research Station, Kearneysville, WV, USA
[6]USDA-ARS, Center for Animal Health and Grain Research, 1515 College Ave, Manhattan, KS 66502, USA
[7]Division of Plant Sciences, University of Missouri, 1-33 Agriculture Building, Columbia MO 65211, USA

CLB, 0000-0002-7013-7083; EGB, 0000-0002-5962-3208;
MH, 0000-0002-6712-9509; TCL, 0000-0002-6299-4384;
WRM, 0000-0002-1663-8741; XJN, 0000-0003-4301-2928

The brown marmorated stink bug, *Halyomorpha halys* (Heteroptera: Pentatomidae), is regarded as one of the world's most pernicious invasive pest species, as it feeds on a wide range of economically important crops. During the autumn dispersal period, *H. halys* ultimately moves to potential overwintering sites, such as human-made structures or trees where it will alight and seek out a final overwintering location, often aggregating with other adults. The cues used during this process are unknown, but may involve vibrational signals. We evaluated whether vibrational signals regulate cluster aggregation in *H. halys* in overwintering site selection. We collected acoustic data for six weeks during the autumn dispersal period and used it to quantify movement and detect vibrational communication of individuals colonizing overwintering shelters. Both movement and vibrational signal production increased after the second week, reaching their maxima in week four, before decaying

again. We found that only males produced vibrations in this context, yet there was no correlation between movement and vibrational signals, which was confirmed through playback experiments. The cues regulating the formation of aggregations remain largely unknown, but vibrations may indicate group size.

## 1. Introduction

Many insect groups, including Hemiptera, use substrate-borne vibrations for short-range communication or for synchronization of behaviours, such as courtship and mating, parent–embryo communication and even egg-hatching [1,2], and it is in hemipterans that mechanical communication is most complex [3]. Other primary functions of plant-borne vibrational signalling in herbivorous social insects are to interact with mutualists, to avoid predation, to exploit food resources through recruitment and to locate a group to join [4,5]. The brown marmorated stink bug, *Halyomorpha halys* (Stål), is a pentatomid bug that uses vibratory signals [6]. Originally from Asia, but now widespread in much of North America and Europe, as well as Chile [7], *H. halys* has the capacity for long-distance flight [8,9], enabling a high dispersal capability. Because of its association with human-modified habitats and its tendency to shelter in enclosed locations, it has become a superb international hitchhiker, making it an important invasive species [7,10]. Attributes that favour its ability to become invasive also include being a long-lived species with high reproductive output [11], extreme polyphagy (including many crops of economic significance) and an overwintering clustering behaviour potentially involving hundreds of tightly packed individuals of both sexes which is enabled by sheltering within human-made structures [7,12,13]. The latter may include suitcases, vehicle openings, sea containers and packaging materials, such as crates, and consequently *H. halys* can easily be transported unnoticed to new locations, making them an ideal invasive species [14].

Before diapause begins, some true bugs, including *H. halys*, aggregate at particular sites, guided by changes in temperature and photoperiod [13,15]. During the autumn dispersal period, insects initially aggregate on exterior locations such as tree trunks, walls [16] and senescent leaves prior to selecting a sheltered overwintering site, usually triggered by the first warm day after the autumn equinox. A key unanswered question is what factors regulate the formation of the initial aggregations prior to dispersal to overwintering sites where adults tightly cluster for the winter diapause [17]. During the autumn dispersal period, *H. halys* chooses sites higher in elevation, as well as ones that are dry and dark [18], while preferred anthropogenic structures on which to alight during autumn dispersal include those that are composed of wood, cement or stone and those that are darker in coloration [19]. However, far less is known about the formation of *H. halys* aggregations within overwintering sites. *Halyomorpha halys* appears to select overwintering sites that are cool, tight, dark and dry [18,20] and, while overwintering, do not respond to their aggregation pheromone [21]. Thus, cues involved in cluster formation in overwintering sites appear not to involve visual or olfactory cues, but perhaps could include vibrational stimuli. This question is of interest to understand the aggregation behaviour of *H. halys* during overwintering site selection as a key aspect of its biology. This may also allow the development of pest control measures that target the aggregation phase, such as attract-and-kill control methods which require that the target species are attracted to a circumscribed location for effective removal from the population [22]. Thus, knowledge of the cues eliciting aggregation behaviour could lead to the development of appropriate methods for control in appropriate areas (e.g. away from storage and cargo activities).

Considerable work has been done on pheromone use by *H. halys* (reviewed in [23]). Baited pheromone traps attract males, females and nymphs [24–26]. However, while a male-emitted aggregation pheromone [27] and a pheromone synergist [28] are known, pheromone traps are poorly suited for managing *H. halys* via mass trapping due to the area of aggregation surrounding baited traps [29] because they may be affected by the crop in which they are deployed [30], or because of additional vibrational cues used during courtship [31]. Based on the fact that during courtship the targeted final approach in pentatomids appears to be mediated by substrate-borne vibrational signals [32,33], Polajnar *et al.* [31] argue that vibrations may be the 'bridge' connecting long-range pheromone attraction and short-range vibrational source localization between conspecifics.

Despite its invasive species status and considerable research effort to prevent invasion into new territories (reviewed in [7]), few studies have investigated vibrational communication in *H. halys*. Using laser vibrometry, Polajnar *et al.* [6] investigated whether vibrational signals play a role in courtship and in aggregating behaviour during summer. For pair formation, they found several male

and female low-frequency (50–80 Hz) signals (three call types for males and two for females) that were highly variable in spectral characteristics and in their combinatorial use. Females responded to male signals on most occasions, but when alone or in pairs, females rarely (ca 10% of trials) produced signals, while males had no such suppression. However, as male signals elicited no attraction from either sex, the authors suggested that male vibratory communication was unlikely to play a major role in aggregation behaviour. Nonetheless, one of the female signals (FS-2) was attractive to males and looked promising for the development of acoustic lures, and later, Mazzoni *et al.* [34] used this call as the basis for the development of an acoustic trap. The authors reported a highly male-biased attraction response to the call, with approximately 50% of males responding, and a corresponding 'loitering' effect around the source for several of these males [34], suggesting some scope for FS-2 to be used as an acoustic trap, but potentially restricted to males. This is similar to the area of aggregation described by Morrison *et al.* [35] around a pheromone source in pheromone traps.

Besides the possible reproductive function of the vibrational signals previously reported [6,34], we hypothesize that vibrational communication is a short-distance cue that regulates cluster formations during autumn dispersal and overwintering site selection. Thus, following calls for further investigation of the function of vibratory signals [34], we aimed to evaluate the role of vibrational communication in *H. halys* cluster formation during the autumn aggregation period prior to dispersal for tight winter clustering. Our goal was to provide new information which may lead to the development of tools for systematically manipulating clusters of individuals. To evaluate this, we recorded the vibrations of *H. halys* movement and substrate-borne vibrational signals during the autumn dispersal period (i.e. autumn in temperate zones; [11]).

# 2. Material and methods

## 2.1. Insect collection

Insect collection was performed during the autumns of 2016 and 2017 in the USA. In 2016, insects were collected during the last stage of the overwintering aggregation process (i.e. late October–early November), and in 2017, insects were collected at the beginning of the aggregation process, just after the autumn equinox (i.e. late September–early October). In 2016, bugs were collected from wooden shelters located in known high-density *H. halys* areas in several counties of West Virginia and Maryland (39°24′50″ N, 78°01′45″ W; 39°30′18″ N, 77°44′35″ W; 39°29′08″ N, 77°46′02″ W; 39°12′28″ N, 77°47′44″ W). In 2017, the insects were obtained from an outdoor colony at the Appalachian Fruit Research Station in Kearneysville, WV (30°21′10″ N, 77°52′37″ W).

## 2.2. Acoustic recordings

This experiment was designed to estimate the correlation between movement and vibratory signal production during the pre-diapause settlement process in *H. halys*. Half of the experiment was conducted in 2016 (late aggregation) and half in 2017 (early aggregation), spanning the full pre-overwintering season. One day before data acquisition, bugs were extracted from the shelters or cages in which they were housed and were sexed and separated into groups of 100 individuals of either all males, all females or a 50/50 mixture of both sexes. After separation, each group was kept in translucent containers under natural light conditions until the following day.

Our set-up was designed to simulate the scenario in which, guided by olfactory cues, *H. halys* begin to aggregate within an enclosed space before forming overwintering clusters. *H. halys* were placed in a 29 cm$^3$ mesh-screened insect cage (made of a plastic frame wrapped in 1 mm$^2$ mesh) which contained an overwintering plywood shelter, as used in previous work [36]. The purpose of the screened cage was to minimize the spatial distribution of the insects in order to accelerate the decision-making process and maximize their time inside the plywood shelter. Each cuboidal plywood shelter (l, w, h: 240 × 190 × 220 mm and 6 mm thick) had an open bottom, which worked as an entry for the bugs, and rested on a 40 mm thick foam square to reduce possible external vibrational artefacts. Additionally, a sloped roof overhung the removable front panel and a 6 mm space between it and the top of the front panel also allowed the insects access to the shelter. Each shelter had 17 layers of cardboard (220 × 180 × 4 mm) and a transducer to measure vibrations was attached on the central (ninth) layer (figure 1a). Layers were separated by two 220 × 30 × 4 mm pieces of cardboard on either side acting as spacers, leaving a 4 mm gap between layers (for a more detailed description of the shelters, see [36]).

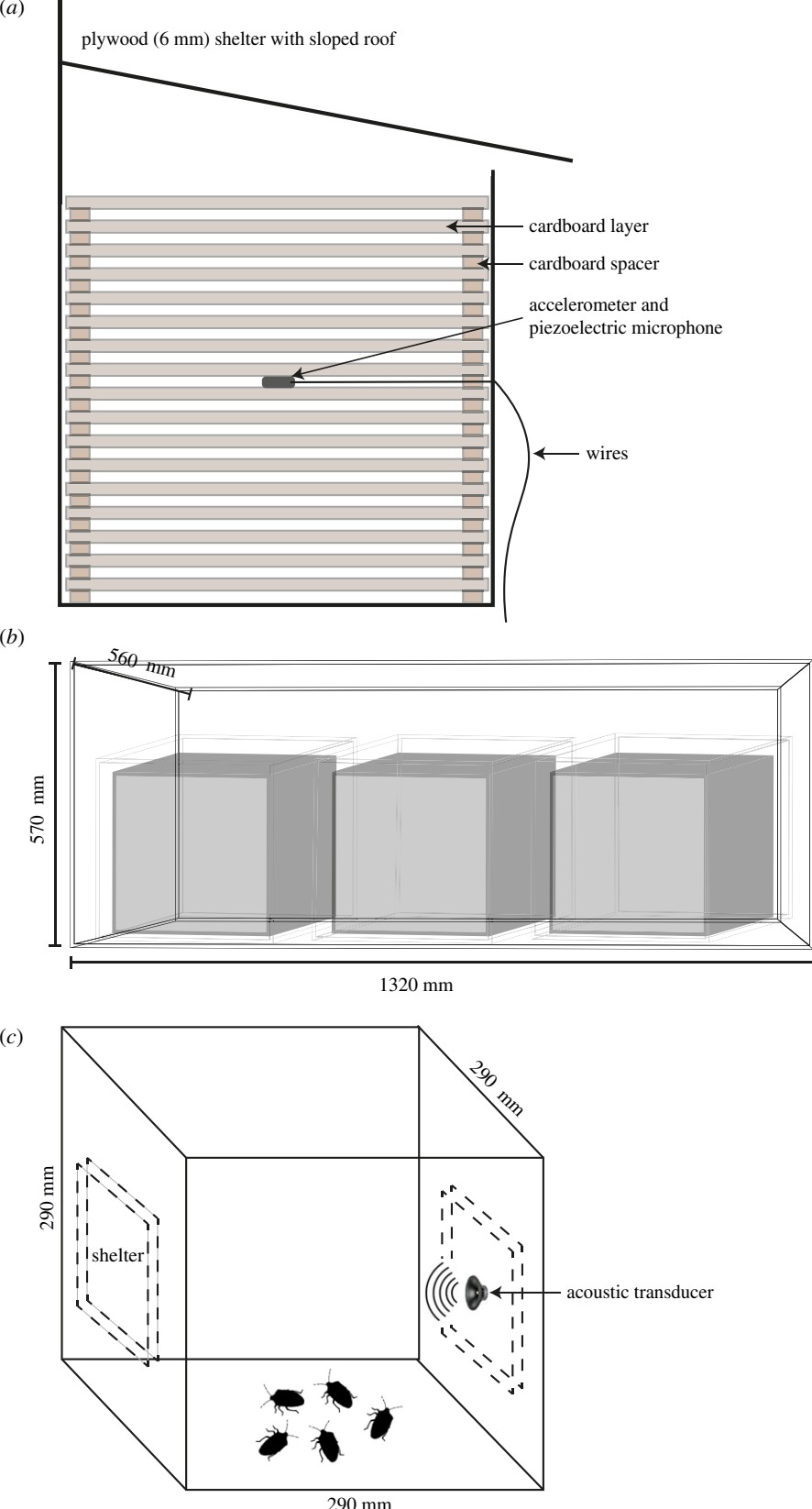

**Figure 1.** Apparatus for acquisition of vibratory signals from *H. halys*. (*a*) Side view of plywood shelter containing 17 layers of cardboard with accelerometer and piezoelectric microphone attached to the ninth layer. (*b*) Soundproof chamber sitting on a foam vibration isolation pad. The chamber contained three shelters, each within a mesh-screen cage. (*c*) Mesh-screen apparatus for behavioural choice tests of five individuals of *H. halys* towards vibrational signals.

Three mesh-screened cages containing the plywood shelters were placed inside a double-glazed soundproof chamber, which was placed on expanded polystyrene to reduce the possibility of external vibrations (figure 1*b*). The experiment consisted in recording the movement and vibrational signals produced by clusters of 100 individuals of *H. halys* (while colonizing an overwintering shelter) over 24 h. At the beginning of each test, each cluster of 100 insects was deployed inside the screened cage which was then closed and data acquisition started. Recordings were acquired with an SD 744T audio recorder (Sound Devices LLC, Reedsburg, WI, USA) and two types of transducer: (i) a piezoelectric microphone (Cold Gold Inc., BC, Canada) and (ii) a 352A24 accelerometer (PCB piezoelectronics, Depew, NY, USA) with a LP201 accelerometer pre-amplifier (Linear X Systems Inc., Tualatin, OR, USA). Since the objective of our experiment was not the description of spectro-temporal features of the signals, but the acoustic detection (i.e. presence/absence) of vibrational signals and the estimation of movement, both types of transducers, after calibration, were equivalent for our intended purposes (electronic supplementary material, figure S1). We also accounted for the transducer effect during the experimental design by alternating the sensors after each replicate for each group of 100 individuals.

For acoustic testing, *H. halys* were deployed in the screened cage in the morning, with the artificial light set to match outdoor lighting conditions. In 2016, we repeated this procedure 10 times for males, females, and the 50/50 mixture (i.e. 30 clusters of 100 individuals) over five weeks. In 2017, we repeated the procedure seven times for each sex group (i.e. 21 clusters of 100 individuals) over three weeks. Overall, we used 5100 insects for this experiment, and each individual was tested only once. After 24 h, the bugs were removed from the set-up, and the shelters, cages and soundproof chamber were cleaned with methanol and hexanes in order to remove possible pheromones from the previous trial.

## 2.3. Acoustic analyses

Movement detection was performed by measuring the average power spectral density (PSD) in the frequency band in which insect movement was observable (500 Hz–5 kHz; figure 3). We calculated the PSD in intervals of 1 min using Welch's method and then averaged the power values occurring between 0.5 and 5 kHz. These values are proportional to the intensity of the sound produced by the movement [37]. By contrast, the lower part of the spectrum (less than 500 Hz), was dedicated solely to the detection of vibrational communication, as this is where *H. halys* vibrational signals occur. We developed a pattern-matching algorithm for the automatic detection of vibrational signals within audio recordings. This consisted of three steps: (i) template generation, (ii) two-dimensional cross-correlation, and (iii) kurtosis estimation (electronic supplementary material, figure S2). Succinctly, the algorithm consisted of generating a 1 min template of the spectrogram of the *H. halys* vibrational signal and then comparing it with the whole dataset partitioned into 1 min sections using two-dimensional cross-correlations. The result of this operation is a two-dimensional matrix, per 1 min section, with information related to the presence–absence of a vibrational signal. We analysed the spectral distribution of values of this matrix (i.e. marginal distribution) and estimated its kurtosis. A detection was considered successful when above-average kurtosis values were found, which means that there was a spike of acoustic activity in that section of the dataset. In order to reduce computational complexity, only the part of the spectrum in which the *H. halys* dominant frequency was located (i.e. 30–100 Hz) was used to detect vibrational signals. The average spectrogram of 15 *H. halys* vibrations (electronic supplementary material, figure S2) was used as template. All analyses were done using Matlab R2018b.

## 2.4. Statistical analyses

After performing the detection of vibrations and the movement estimation, we analysed the stimulus-response asynchrony to test whether the stimulus (i.e. vibrational signal) triggered the hypothesized response (i.e. movement). We compared the changes in activity before and after a vibration was produced in time intervals (lags) of 1, 5 and 10 min. Changes in the movement were estimated using $\Delta m^- = |p_{L1} - p_n|$ and $\Delta m^+ = |p_{L2} - p_n|$, where $\Delta m^-$ and $\Delta m^+$ are the movement differentials before and after the stimulus has occurred, $p_n$ is the average power value in the 0.5–5 kHz frequency band at the instant $n$ of the stimulus (i.e. frequency band used for movement estimation), and $p_{L1}$ and $p_{L2}$ are the averaged movement power values for each time interval before and after the stimulus. As the movement differentials were continuous and normally distributed, we used a paired *t*-test to compare them before and after a vibration occurred. We also used Pearson's correlation coefficient to determine

the linear correlation of the movement differentials before and after a vibration and compared the data using polynomial models. *Halyomorpha halys* individuals were considered moving when the recorded movement amplitude values were above the DC offset. Using this information, the movement was discretized (1–0) and compared to the presence–absence of vibrational communication. The level of correlation between the discretized movement and the presence of vibrational communication was estimated using Pearson's phi coefficient ($\varphi$). Weekly changes in vibrational signal production were analysed using Pearson's chi-squared test ($\chi^2$). *Post hoc* comparisons were estimated using additional Bonferroni-corrected $\chi^2$ tests for all factor combinations. Changes in movement patterns across the season were analysed using analysis of variance (ANOVA), after testing for normality and homoscedasticity using Shapiro–Wilk's and Levene's tests, respectively, and Tukey's HSD test was used for *post hoc* comparisons. As data from behavioural experiments (see below) was not normally distributed, we used Wilcoxon signed-rank tests for non-independent samples in GraphPad Prism8 to analyse choices made, excluding individuals that did not choose a shelter (undecided).

## 2.5. Behavioural experiments

We performed a series of playback experiments (sample sizes indicated in results below) to determine whether *H. halys* vibrational signals are able to attract other individuals. Our experimental protocol consisted of placing five *H. halys* in the middle of a screened cage, as used in the acoustic recording tests, and giving them the option of colonizing one of two shelters over the course of 24 h (figure 1*c*). After 24 h, the sex and the number of individuals in each shelter were recorded and the individuals were removed.

Before inserting them into the screened cage, insects were placed inside a Petri dish on ice to minimize erratic movement. Then, the Petri dish was gently placed in the middle of the box from where the bugs dispersed toward the shelters. Shelters were made of two of the cardboard layers of the wooden boxes used in the acoustic experiment and were located on opposite sides of the cage. One of the shelters had an attached acoustic transducer (GHXamp 40 mm speaker, 20 Hz–20 kHz frequency response) which was moved before each trial to account for laterality effects. Between trials, the cage and shelters were cleaned with methanol and hexanes to remove possible pheromones from the previous trial. No individual was ever used more than once. Experiments were performed under artificial light conditions matching the photoperiod that induces diapause behaviour (L : D 10 : 14) [38]. The amplitude of the playback recordings at the source (expressed as substrate displacement velocity), both on the transducer and on the surface of the shelter, was 152.5 and 1.59 mm s$^{-1}$, respectively. The amplitude at the middle of the cage (release point) was 0.77 mm s$^{-1}$. Amplitude measurements were acquired using a OFV-512 laser vibrometer (Polytec, Waldbronn, Germany) with a OFV-153 reference head (Polytec) connected to a OFV-5000 vibrometer controller (Polytec) and a DL7440 oscilloscope (Yokogawa, Tokyo, Japan).

In the first experiment, each group of five *H. halys* consisted of either three females and two males, or two males and three females. Each group was given the choice between one shelter with *H. halys* pre-diapause vibrations and one without vibrations (silent). Here, the transducer was set to randomly play calls of 10 different males (these varied between 60 and 90 s each) on a loop with a 20 s inter-stimulus interval between calls. The other shelter had no calls presented. To determine if *H. halys* was attracted to the vibrational signals, we counted how many individuals went to each shelter, irrespective of sex. Additionally, to determine if there were sex differences toward the vibrational signals, we considered the numbers of each sex that went toward each shelter.

As we also wanted to test if *H. halys* aggregated to any low-frequency noise, or specifically to male vibrational signals, we performed a second experiment using the same set-up and experimental conditions, but testing brown noise (PSD~$1/f^2$) instead of *H. halys* vibrations.

## 3. Results

We were able to record mechanical vibrations caused by the movement of *H. halys* within the shelters. These vibrations were detected between 500 Hz and 5 kHz (electronic supplementary material, figure S3). Additionally, males, but not females, produced acoustically detectable signalling vibrations (figure 2), which were very distinct from the movement-based vibrations detected. These signals had a duration of $14.52 \pm 6.09$ s ($n = 114$, mean ± s.d.), characterized by sporadic downward frequency-modulations between $54.4 \pm 2.2$ and $85.7 \pm 4.9$ Hz ($n = 114$) and a harmonic content below

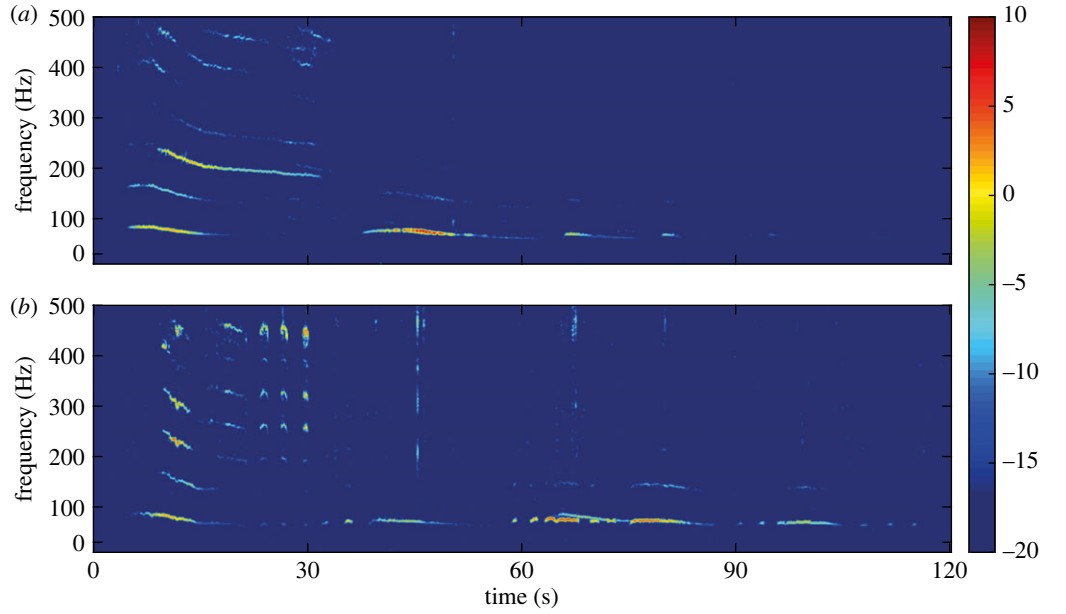

**Figure 2.** Spectrograms depicting the two types of vibrations (long and short) produced by male *H. halys*. These vibrations were acquired in shelters containing only males. (*a*) Signals generated by a single individual. (*b*) Two males vibrating simultaneously; second male starts at *ca* 60 s. Male vibrational behaviour often lasted several minutes, whereas females did not produce vibratory signals. Colourbar in dB.

500 Hz, similar to the male song 1 (MS-1) reported by Polajnar *et al.* [6]. In less than a fifth of cases, these vibrations were followed by shorter pulses of $1.13 \pm 0.41$ s duration ($n = 87$) with spectral distribution between $56.6 \pm 2.0$ and $77.4 \pm 7.5$ Hz ($n = 87$) (figure 2), akin to the MS-2 signals previously reported [6].

Most vibrational signalling activity occurred during daytime (68.9%), particularly between 12.00 and 17.00 (54.8%; figure 3). We then compared the week-on-week trends for both movement and vibrational signals. Movement changed significantly through the autumn dispersal season ($F_{5, 8634} = 131.4$, $p < 0.001$). Specifically, the mean movement in the third week (late October) was significantly higher (M = 0.005, s.d. = 0.007) than the first (M = 0.0005, s.d. = 0.0032) and second week (M = 0.0004, s.d. = 0.0042). After this, activity started decreasing until the beginning of the diapause tight clustering stage (figure 3). A similar pattern was found for vibrational communication ($\chi_5^2 = 127.9$, $p < 0.001$, $n = 8640$), where signal production spiked after the second week ($\chi^2 = 116.6$, $p < 0.001$, $n = 1440$) and then monotonically decreased (figure 3). As seen in figure 3, there were clusters of spikes in weeks 3–6. These may have been the same male calling repeatedly, or may have been separate males, possibly calling in response to detected signals. Simultaneous signal production between three or more individuals was never observed.

Our correlation analysis suggested that movement during the autumn dispersal period was not triggered by vibratory signals ($\varphi = -0.099$, $p < 0.001$). In fact, 83.9% of the vibrational communication during the six weeks took place when the individuals were inactive (figure 4). To corroborate this, we compared the changes in activity before and after a vibration was produced in lags of 1, 5 and 10 min (table 1). There were no significant changes in the movement for any of the tested conditions (i.e. all male and 50/50 mixture, as females did not produce vibrations; table 1). The data were also highly correlated ($\rho > 0.77$ for all conditions, table 1). This was corroborated by the linear models, where all approached $y \approx \alpha x$ ($r^2 > 0.99$, table 1), meaning that the stimulus (i.e. vibration) did not have any effect on the response (i.e. movement).

For mixed-sex groups of five *H. halys*, there was no preference for vibrational signals over no-vibration controls (W = 24, $p = 0.362$, $n = 32$), and this mirrored our results using brown noise versus no-vibration controls (W = 20, $p = 0.547$, $n = 36$; figure 5). Of the insects that moved to shelters in vibrational signal playback tests, males 12/29 (41%) moved toward the shelter playing the vibration slightly less frequently than females 16/24 (67%) (Fisher exact test $p = 0.098$). This suggests there was no tendency to colonize spaces emitting any kind of vibration. Insects also split into groups of different sizes going into each shelter, ruling out the following of pheromone trails.

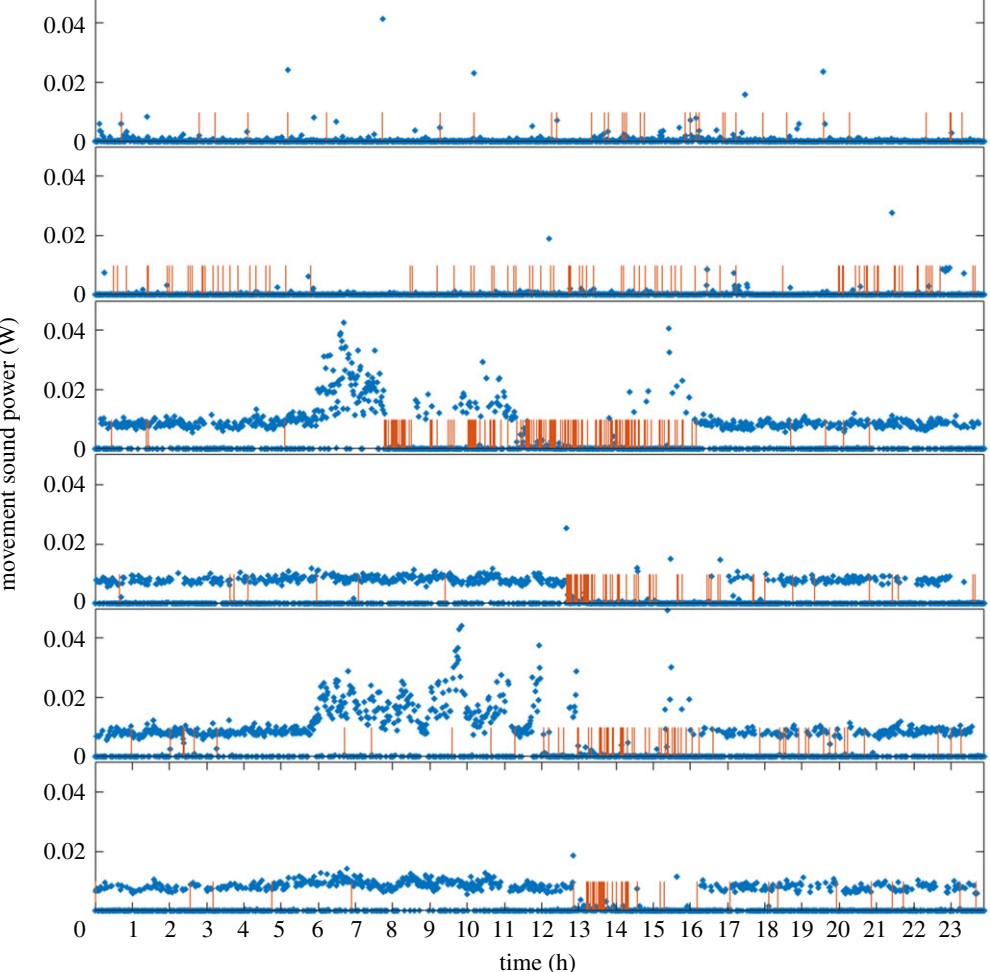

**Figure 3.** Sound power produced by the movement of 100 *H. halys* males contained in a wooden shelter over a 24 h time span over six weeks during the pre-diapause season (October–mid-November), where week 1 is the top panel and week 6 is the bottom panel. Spikes represent the time of day where calling vibrations were detected and tended to reach peak levels during periods in which insects were least active.

## 4. Discussion

We found no vibrational signals in female-only set-ups, consistent with previous work [6]. However, male *H. halys* do produce vibrations during the autumn dispersal period, and there is a clear pattern of this increasing over the six-week period, before ramping down again. We also found that vibrational signalling activity tended to occur between 12.00 and 17.00, which correlates with their peak period for alighting on walls during the autumn dispersal [16], possibly due to light levels at that time of year [39]. However, we found that vibrational signalling increased during periods of the day when movement within the shelters decreased, which is a common pattern among insects that use vibrational communication [4]. In contrast with our predictions, our results suggest that the function of these signals is not critical to locating conspecifics and eliciting aggregation behaviour over short distances. Thus, substrate-borne signalling does not appear to be a causative factor in cluster formation during the autumn dispersal period. Clustering may instead be regulated by abiotic factors, such as gradients in temperature, humidity or light conditions among microclimates (e.g. [39]), perhaps coupled with some self-aggregation social rule we have not yet considered. As these insects were already in the laboratory prior to testing, it is conceivable that we somehow interrupted their clustering behaviour when we placed them into our arenas, but this does not explain the continued signalling and movement that we observed and has not been a factor disrupting other behaviours in this species (e.g. [6,34]). Additionally, we found no cases of mating pairs inside the shelters. We also found that signalling increased during the times in which movement decreased. This inverse

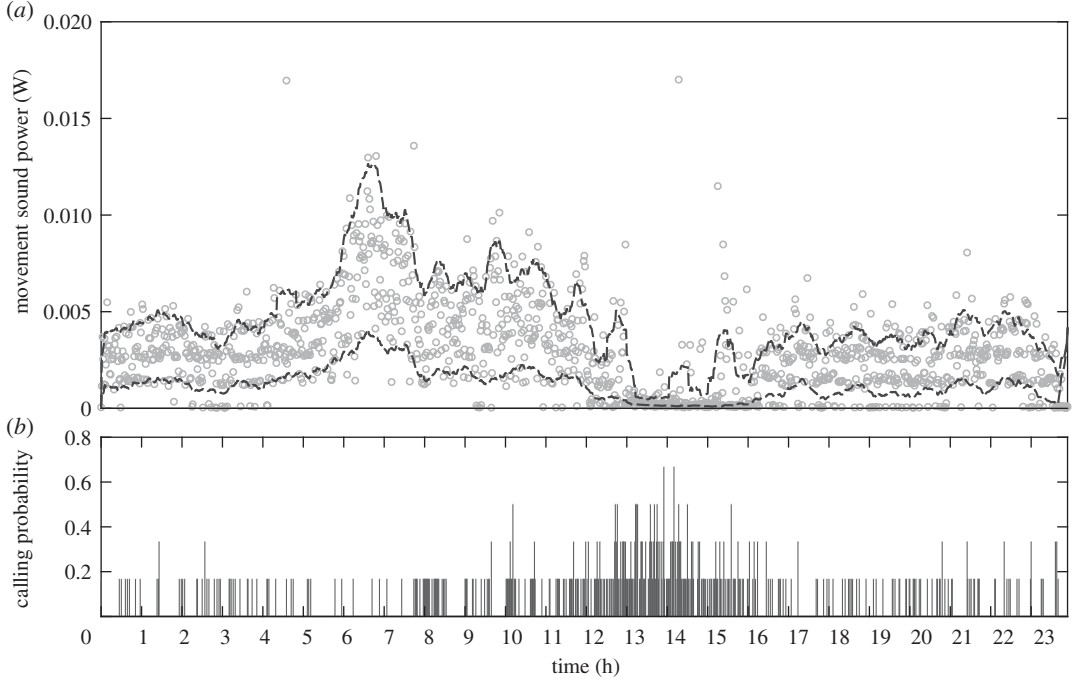

**Figure 4.** Average sound power from the movement (*a*) and calling probability (*b*) of *H. halys* males contained in wooden shelters during a 24 h time span. Dots represent the average sound amplitude of movement in 1 min time bins. Dotted lines indicate SEM. Most vibrational activity occurred from 12.00 to 17.00, which was also the most inactive period.

**Table 1.** Linear models ($y = \alpha x + \beta$), Pearson's correlation coefficients ($\rho$) and paired *t*-tests (*t, df, p*) for all the movement differentials before and after a *H. halys* vibration inside a wooden shelter. Movement differentials were estimated using three different time intervals (1, 5 and 10 min) for two conditions (all males, 50/50 mixture of males and females). $p < 0.001$ for all correlation coefficients.

| condition | lag | $\alpha$ (95% CI) | $\beta$ (95% CI)*1 e-05 | $r^2$ | $\rho$ | *t* | *df* | *p* |
|---|---|---|---|---|---|---|---|---|
| 100 ♂ | 1 | 0.980 (0.976, 0.985) | 0.909 (−4.612, 6.432) | 0.9967 | 0.774 | 0.461 | 476 | 0.64 |
| 100 ♂ | 5 | 0.995 (0.992, 0.997) | 0.397 (−2.283, 3.077) | 0.9992 | 0.948 | 0.346 | 476 | 0.72 |
| 100 ♂ | 10 | 0.997 (0.994, 0.999) | −0.675 (−3.004, 1.653) | 0.9994 | 0.964 | 0.399 | 476 | 0.69 |
| 50♂/50♀ | 1 | 0.195 (0.193, 0.197) | 14.23 (10.26, 18.20) | 0.9978 | 0.790 | 0.692 | 247 | 0.48 |
| 50♂/50♀ | 5 | 0.963 (0.962, 0.964) | 3.604 (2.387, 4.820) | 0.9998 | 0.974 | 0.492 | 247 | 0.62 |
| 50♂/50♀ | 10 | 0.982 (0.981, 0.983) | 1.685 (−2.970, 3.668) | 0.9995 | 0.987 | 1.144 | 247 | 0.25 |

relationship between signalling and movement suggests that these signals do not function in an anti-predator context, as signalling in this context also typically elicits aggregating behaviour [4] and would thus trigger movement. Our experimental procedure also largely excluded recruitment for foraging purposes as a function for any detected signals. While a function for these signals remains to be determined, we found no mating couples, and it is noteworthy that only males called during this extended autumn period and it is unlikely to be related to aggregation, as there is no known overwintering pheromone in this species—*H. halys* tend to stop producing the aggregation pheromone as they disperse, and stop responding to the aggregation pheromone over winter [21]. Our results suggest that whatever the function of these vibrational signals, it is male-driven and may be somewhat 'group-based' or territorial, possibly giving others an indication of whether an aggregation has reached a critical size. This hypothesis may match the observed patterns in which signalling is relatively low for the first few weeks, when *H. halys* disperses to find an overwintering location, and attenuates at the end of the autumn period, when the insects are likely to have already found a place to cluster.

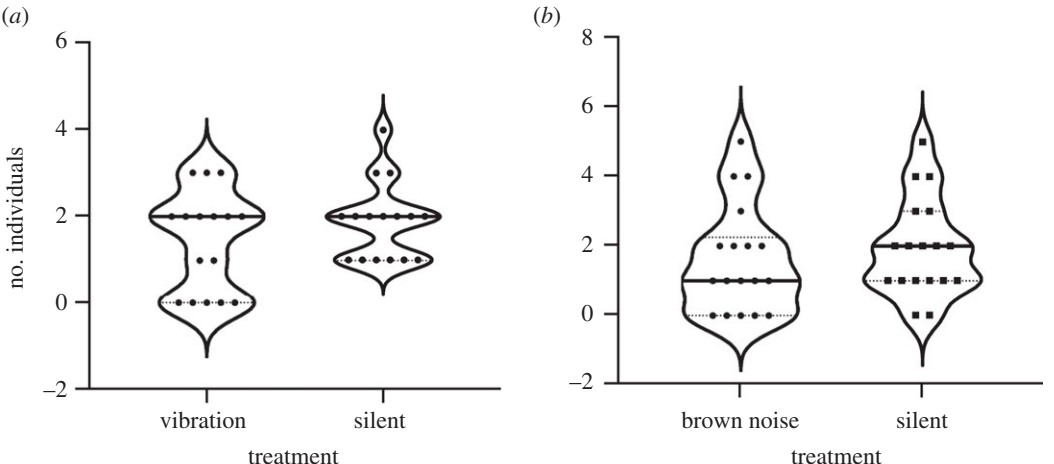

**Figure 5.** Violin plots depicting median (black line), quartiles (dotted lines) and raw data points from 24 h playback experiments testing phonotaxis of *H. halys* toward (*a*) male vibration and (*b*) brown noise in mixed-sex groups of five individuals.

There are a few alternative explanations that may explain clustering prior to dispersal in autumn. It is possible that *H. halys* may seek physical contact with each other during dispersal, and movement into an aggregation is halted when they perceive the hydrocarbons of conspecifics. For example, other non-social arthropods may react to the hydrocarbons of conspecifics in disparate ways, including both behavioural attraction or aversion [40]. After dispersal to a final overwintering site, *H. halys* responds to thigmotactic cues, based on antennal contact [17], to initiate tight cluster formation (even when in the presence of freshly dead insects [41]), and a similar process may happen prior to autumn dispersal. Another possibility is that there may be some as-yet-to-be-determined trail-following mark or pheromone used by *H. halys* during autumn clustering. During the period in which we carried out these experiments, we regularly witnessed *H. halys* flying from afar onto a tree and taking the same or very similar routes to other *H. halys* that had previously arrived on that tree. Such a mark could be deposited through the tarsi; this may not be an unreasonable hypothesis given the fact that the wasp *Trissolcus japonicus* (Ashmead) (Hymenoptera: Scelionidae), the main parasitoid of *H. halys*, strongly responds, by stopping or significantly slowing down movement when on leaf surfaces containing *H. halys* tarsal prints [42]. Ensuing work should evaluate these alternative explanations for the formation of autumn aggregations by *H. halys* prior to dispersal.

Polajnar *et al.* [6] found long pulses (e.g. MS-1) were spontaneously emitted in male–male interactions. However, their behavioural experiments showed that these male signals attracted neither males nor females. Since no function was found, Polajnar *et al.* [6] hypothesized that these pulses could have a function in contexts other than reproduction. The MS-1 pulses have similar characteristics to the ones found during our experiment, which generally consist of a single acoustic unit, are sporadically emitted and can be found in male–male interactions. In spite of the similarities between the male calls previously reported [6] and the ones found here, we were unable to decipher the function of these calls. Although we found that males of *H. halys* emit vibratory signals during the autumn dispersal period and overwintering site selection processes, these vibrations do not trigger movement in conspecifics, indicating that vibrational communication is probably not one of the modes that regulates close-range aggregations in this species. Polajnar *et al.* [6] performed their work during the spring and summer, when overwintering aggregation behaviour is absent. By performing our experiments during a different time of the year, at a unique phase in the biology of *H. halys* (i.e. during the autumn dispersal phase), we thought that we might be able to elucidate the function of these *H. halys* male vibrations. Despite not determining a function for this call, we can confidently state that short-distance pre-diapause aggregation is not one of its possible functions.

Ethics. This work follows the Association for the Study of Animal Behaviour/Animal Behavior Society guidelines for the use of animals in research, and complies with institutional and national guidelines.
Data accessibility. All data are provided in the electronic supplementary material.
Authors' contributions. C.L.B. and M.H. performed the analyses. C.L.B. performed the experiments and C.L.B., E.G.B, T.C.L, W.R.M. K.B.R. and X.J.N. wrote the manuscript.
Competing interests. We declare we have no competing interests.

Funding. This work was supported by the New Zealand Ministry of Business, Innovation, and Employment (C04X1407) and B3 funding to Scion. The use of trade names is for the purposes of providing scientific information only and does not constitute endorsement by the United States Department of Agriculture. The USDA is an equal opportunity employer.

Acknowledgements. We are grateful to Sharon Jones for administrative help and to Torri Hancock, Nicole Davidson, Scott Wolford and Amy Tabb for technical help.

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
