## [Reviewer comments · Royal Society Open Science]

Review History

RSOS-201371.R0 (Original submission)

Review form: Reviewer 1

Is the manuscript scientifically sound in its present form?

No

Are the interpretations and conclusions justified by the results?

No

Is the language acceptable?

Yes

Do you have any ethical concerns with this paper?

No

Have you any concerns about statistical analyses in this paper?

Yes

Recommendation?

Major revision is needed (please make suggestions in comments)

Comments to the Author(s)

This paper examined the effects of male vibrations on forming aggregation in overwintering site selection in *Halyomorpha halys*. The authors found no significant changes in movement between before and after a vibration was produced. The authors furthermore confirmed the results by playback experiments. They concluded that the overwintering aggregations were not regulated by male vibrations in this species with elaborate and thoroughly replicated experiments. The findings are important in understanding the function of vibrational signals in this species, and my overall feeling is that the study is worth of publication. However, there are some concerns mainly in statistics shown below which should be addressed before publication.

L 177: Why did the authors use Pearson's correlation? In my understanding, the hypothesis is that the activity level varied between before and after vibrations were produced. However, the authors seem to test if the activity level after vibrations were produced was related to that before vibrations were produced. Similar type of problems may occur in other parts (e.g., line 184). Paired t test or mixed models should be used for the paired samples.

L 190: The authors used GLMs, but data in the sample are not independent. For example, within a trial, the number of bugs in one shelter is probably related to that in the other shelter. Statistical models for paired samples should be used again.

L 195: Did the author check the level of the vibrations played? The information on amplitude of played vibrations should be provided.

According to Fig 6, the vibratory spikes clearly occur in a clustered manner in week 3-6. Do males produce vibrations in response to vibratory signals produced by other males?

Some figures (e.g., Fig 2-4) may be moved to supplementary data.

Review form: Reviewer 2

Is the manuscript scientifically sound in its present form?

No

Are the interpretations and conclusions justified by the results?

Yes

Is the language acceptable?

Yes

Do you have any ethical concerns with this paper?

No

Have you any concerns about statistical analyses in this paper?

No

Recommendation?

Accept with minor revision (please list in comments)

Comments to the Author(s)

An interesting report by Bedoya et al., in essence testing a hypothesis about potential role of spontaneous vibrational emissions by BMSB males in establishing autumn aggregations. Even though the results do not support the hypothesis, I believe such a negative result would be a valuable contribution to the field and an incentive for researchers to look for real mechanisms regulating aggregation behaviour of this important pest, as well as the real role of MS-1 signals. To a lesser extent, the report is also valuable as a partial replication of recordings by Polajnar et al. (2016), confirming similar vibratory behaviour of an American population in a different season, although this aspect requires improvement (see below).

Results lack more precise descriptions of recorded signals (L224-230); all temporal and frequency ranges should include basic descriptive statistics, terms like "occasionally", "usually" and "on rare occasions" must be avoided.

More importantly, playback experiments are not reproducible from this report which lacks information about playback amplitude, at least at the source, but preferably at various points within the experimental arena - both in absolute terms and relative to signals emitted by live males. Amplitude could importantly affect attractiveness of artificial signals, either negatively or positively. Luckily, the issue is probably easy to resolve by simple calibrated measurement. L205 needs precise information about the acoustic transducer used, and the report would benefit from including a sample stimulation sequence and a length of brown noise as supplementary sound files.

Additional minor comments:

L151-153: Which software for acoustic measurements?

L167: Please state exactly which "vibrations" were averaged. Consider including the average spectrogram as a figure.

L181: Be careful about terminology - in a behavioural study, the word "signal" should be reserved for vibratory emissions used in communication to avoid confusion.

L212-213: I assume "calls" mean MS1+MS2 (judging by duration longer than MS1 and by what is shown in Fig. 5), but please specify what exactly was played, preferably with a sound file included as a supplement.

L256: "Ruling out" is perhaps too strong a statement, given that the authors didn't study this in detail.

L279-280: Any comment about possible association with aggregation pheromone production, which is also male-only and may indicate group size? A statement about whether you found any mating couples could also be informative.

L288-290: Discussion about sensory cues for cluster formation should mention the work of Toyama et al. (2006).

Decision letter (RSOS-201371.R0)

Dear Dr Bedoya

The Editors assigned to your paper RSOS-201371 "Brown marmorated stink bug overwintering aggregations are not regulated through vibrational signals during autumn dispersal" have now received comments from reviewers and would like you to revise the paper in accordance with the

reviewer comments and any comments from the Editors. Please note this decision does not guarantee eventual acceptance.

Please submit your revised manuscript and required files (see below) no later than 21 days from today's (ie 02-Oct-2020) date. Note: the ScholarOne system will 'lock' if submission of the revision is attempted 21 or more days after the deadline. If you do not think you will be able to meet this deadline please contact the editorial office immediately.

on behalf of Dr Krijn Paaijmans (Associate Editor) and Pete Smith (Subject Editor)
openscience@royalsociety.org

Reviewer comments to Author:

Reviewer: 1
Comments to the Author(s)

This paper examined the effects of male vibrations on forming aggregation in overwintering site selection in *Halyomorpha halys*. The authors found no significant changes in movement between before and after a vibration was produced. The authors furthermore confirmed the results by playback experiments. They concluded that the overwintering aggregations were not regulated by male vibrations in this species with elaborate and thoroughly replicated experiments. The findings are important in understanding the function of vibrational signals in this species, and my overall feeling is that the study is worth of publication. However, there are some concerns mainly in statistics shown below which should be addressed before publication.

L 177: Why did the authors use Pearson's correlation? In my understanding, the hypothesis is that the activity level varied between before and after vibrations were produced. However, the authors seem to test if the activity level after vibrations were produced was related to that before

vibrations were produced. Similar type of problems may occur in other parts (e.g., line 184). Paired t test or mixed models should be used for the paired samples.

L 190: The authors used GLMs, but data in the sample are not independent. For example, within a trial, the number of bugs in one shelter is probably related to that in the other shelter. Statistical models for paired samples should be used again.

L 195: Did the author check the level of the vibrations played? The information on amplitude of played vibrations should be provided.

According to Fig 6, the vibratory spikes clearly occur in a clustered manner in week 3-6. Do males produce vibrations in response to vibratory signals produced by other males?

Some figures (e.g., Fig 2-4) may be moved to supplementary data.

Reviewer: 2

Comments to the Author(s)

An interesting report by Bedoya et al., in essence testing a hypothesis about potential role of spontaneous vibrational emissions by BMSB males in establishing autumn aggregations. Even though the results do not support the hypothesis, I believe such a negative result would be a valuable contribution to the field and an incentive for researchers to look for real mechanisms regulating aggregation behaviour of this important pest, as well as the real role of MS-1 signals. To a lesser extent, the report is also valuable as a partial replication of recordings by Polajnar et al. (2016), confirming similar vibratory behaviour of an American population in a different season, although this aspect requires improvement (see below).

Results lack more precise descriptions of recorded signals (L224-230); all temporal and frequency ranges should include basic descriptive statistics, terms like "occasionally", "usually" and "on rare occasions" must be avoided.

More importantly, playback experiments are not reproducible from this report which lacks information about playback amplitude, at least at the source, but preferably at various points within the experimental arena - both in absolute terms and relative to signals emitted by live males. Amplitude could importantly affect attractiveness of artificial signals, either negatively or positively. Luckily, the issue is probably easy to resolve by simple calibrated measurement. L205 needs precise information about the acoustic transducer used, and the report would benefit from including a sample stimulation sequence and a length of brown noise as supplementary sound files.

Additional minor comments:

L151-153: Which software for acoustic measurements?

L167: Please state exactly which "vibrations" were averaged. Consider including the average spectrogram as a figure.

L181: Be careful about terminology - in a behavioural study, the word "signal" should be reserved for vibratory emissions used in communication to avoid confusion.

L212-213: I assume "calls" mean MS1+MS2 (judging by duration longer than MS1 and by what is shown in Fig. 5), but please specify what exactly was played, preferably with a sound file included as a supplement.

L256: "Ruling out" is perhaps too strong a statement, given that the authors didn't study this in detail.

L279-280: Any comment about possible association with aggregation pheromone production, which is also male-only and may indicate group size? A statement about whether you found any mating couples could also be informative.

L288-290: Discussion about sensory cues for cluster formation should mention the work of Toyama et al. (2006).

===PREPARING YOUR MANUSCRIPT===

- one version identifying all the changes that have been made (for instance, in coloured highlight, in bold text, or tracked changes);
- a 'clean' version of the new manuscript that incorporates the changes made, but does not highlight them. This version will be used for typesetting if your manuscript is accepted.

===PREPARING YOUR REVISION IN SCHOLARONE===

<https://royalsociety.org/journals/authors/author-guidelines/#supplementary-material> to include a suitable title and informative caption. An example of appropriate titling and captioning may be found at https://figshare.com/articles/Table_S2_from_Is_there_a_trade-off_between_peak_performance_and_performance_breadth_across_temperatures_for_aerobic_sc_ope_in_teleost_fishes_/3843624.

Author's Response to Decision Letter for (RSOS-201371.R0)

See Appendix A.

RSOS-201371.R1 (Revision)

Review form: Reviewer 1

Is the manuscript scientifically sound in its present form?

Yes

Are the interpretations and conclusions justified by the results?

Yes

Is the language acceptable?

Yes

Do you have any ethical concerns with this paper?

No

Have you any concerns about statistical analyses in this paper?

No

Recommendation?

Accept as is

Comments to the Author(s)

I found the authors have answered the queries appropriately and am happy to accept this interesting manuscript.

Review form: Reviewer 2

Is the manuscript scientifically sound in its present form?

Yes

Are the interpretations and conclusions justified by the results?

Yes

Is the language acceptable?

Yes

Do you have any ethical concerns with this paper?

No

Have you any concerns about statistical analyses in this paper?

No

Recommendation?

Accept with minor revision (please list in comments)

Comments to the Author(s)

I am satisfied with the authors' response to my and the first reviewer's comments, good work. Some very minor issues remain (see below), but they do not diminish the scholarly quality of this manuscript.

Minor issues:

L231-L239: n values aren't reported with descriptive statistics

Occasionally awkward prose:

"Additionally, our experimental procedure also largely excluded..." (L288)

"This suggests that..." (L286, L293)

Decision letter (RSOS-201371.R1)

Dear Dr Bedoya

On behalf of the Editors, we are pleased to inform you that your Manuscript RSOS-201371.R1 "Brown marmorated stink bug overwintering aggregations are not regulated through vibrational signals during autumn dispersal" has been accepted for publication in Royal Society Open Science subject to minor revision in accordance with the referees' reports. Please find the referees' comments along with any feedback from the Editors below my signature.

Please submit your revised manuscript and required files (see below) no later than 7 days from today's (ie 29-Oct-2020) date. Note: the ScholarOne system will 'lock' if submission of the revision is attempted 7 or more days after the deadline. If you do not think you will be able to meet this deadline please contact the editorial office immediately.

Kind regards,

Anita Kristiansen
Editorial Coordinator

on behalf of Dr Krijn Paaijmans (Associate Editor) and Pete Smith (Subject Editor)
openscience@royalsociety.org

Reviewer comments to Author:
Reviewer: 2

Comments to the Author(s)

I am satisfied with the authors' response to my and the first reviewer's comments, good work. Some very minor issues remain (see below), but they do not diminish the scholarly quality of this manuscript.

Minor issues:

L231-L239: n values aren't reported with descriptive statistics

Occasionally awkward prose:

"Additionally, our experimental procedure also largely excluded..." (L288)

"This suggests that..." (L286, L293)

Reviewer: 1

Comments to the Author(s)

I found the authors have answered the queries appropriately and am happy to accept this interesting manuscript.

===PREPARING YOUR MANUSCRIPT===

- one version identifying all the changes that have been made (for instance, in coloured highlight, in bold text, or tracked changes);
- a 'clean' version of the new manuscript that incorporates the changes made, but does not highlight them. This version will be used for typesetting.

===PREPARING YOUR REVISION IN SCHOLARONE===

-- If you have uploaded ESM files, please ensure you follow the guidance at <https://royalsociety.org/journals/authors/author-guidelines/#supplementary-material> to include a suitable title and informative caption. An example of appropriate titling and captioning may be found at https://figshare.com/articles/Table_S2_from_Is_there_a_trade-off_between_peak_performance_and_performance_breadth_across_temperatures_for_aerobic_scops_in_teleost_fishes_/3843624.

Author's Response to Decision Letter for (RSOS-201371.R1)

See Appendix B.

Decision letter (RSOS-201371.R2)

Dear Dr Bedoya,

It is a pleasure to accept your manuscript entitled "Brown marmorated stink bug overwintering aggregations are not regulated through vibrational signals during autumn dispersal" in its current form for publication in Royal Society Open Science.

on behalf of Dr Krijn Paaijmans (Associate Editor) and Pete Smith (Subject Editor)
openscience@royalsociety.org

Appendix A

Response to the Reviewers

We would like to thank the reviewers for the time and effort they put into reviewing our manuscript. We appreciate their positive outlook on our manuscript and the constructive feedback and think the manuscript has significantly improved as a consequence the suggested changes. We have addressed all concerns raised by the reviewers, and accepted all their recommendations. Below, we present our responses to each of the comments in order. Line numbers in our responses refer to the line numbers in the 'clean' version of the revised manuscript.

REVIEWER # 1

This paper examined the effects of male vibrations on forming aggregation in overwintering site selection in *Halyomorpha halys*. The authors found no significant changes in movement between before and after a vibration was produced. The authors furthermore confirmed the results by playback experiments. They concluded that the overwintering aggregations were not regulated by male vibrations in this species with elaborate and thoroughly replicated experiments. The findings are important in understanding the function of vibrational signals in this species, and my overall feeling is that the study is worth of publication. However, there are some concerns mainly in statistics shown below which should be addressed before publication.

Q1: L 177: Why did the authors use Pearson's correlation? In my understanding, the hypothesis is that the activity level varied between before and after vibrations were produced. However, the authors seem to test if the activity level after vibrations were produced was related to that before vibrations were produced. Similar type of problems may occur in other parts (e.g., line 184). Paired t test or mixed models should be used for the paired samples.

R1// Great observation! We have now added the analyses suggested by the reviewer (paired t-tests) as a complement to our analyses in the manuscript (Lines 179-181, 257, and table 1). We used the Pearson's correlation coefficient because changes in movement can naturally occur, so it was important to show that movement before and after a vibration was in fact triggered by the vibration and thus uncorrelated. Due to the nature of our work, we decided to present linear models and correlation coefficients because we thought they demonstrated in a more direct manner our results. Nonetheless, we think the added analyses suggested by the reviewer are also needed and improve the quality of our manuscript.

Q2: L 190: The authors used GLMs, but data in the sample are not independent. For example, within a trial, the number of bugs in one shelter is probably related to that in the other shelter. Statistical models for paired samples should be used again.

R2// This is a very valid point, thank you. As the data are non-parametric, we have reanalysed these data using Wilcoxon signed rank tests for non-independent samples. The results are the same (Lines 192-194 and 261-267). As a consequence, we have changed our plot (now figure 5) to violin plots depicting median, quartiles and raw data, and have removed table 2 (GLM model results).

Q3: L 195: Did the author check the level of the vibrations played? The information on amplitude of played vibrations should be provided.

R3// Thank you for pointing out this omission. Yes, amplitude levels of the signals were checked with an accelerometer before performing the playback experiments. We have now used a laser vibrometer to report the substrate velocity displacement at the source (both on the transducer and on the shelter) and at the release point (middle of the cage) (Lines 211-217).

Q4: According to Fig 6, the vibratory spikes clearly occur in a clustered manner in week 3-6. Do males produce vibrations in response to vibratory signals produced by other males?

R4// Interesting question. Based on the spectrograms, usually it appeared to be a single male that called on multiple instances (the tightest sequential calling by the same male was in about 60 s intervals (see figure below). We rarely detected the simultaneous calls of two males, and never more than two. Signal production is not tightly clustered, as seen from an enlargement of a portion of figure 6, depicting one hour (13-14:00) in weeks 3,4,5 and 6. The tightest clustering of vibratory spikes (sampled at 1 min intervals) typically lasted only a few minutes (<10 min). We have added some information pertaining to this in our results (lines 248-251).

Q5: Some figures (e.g., Fig 2-4) may be moved to supplementary data.

R5// Figures 2, 3, and 4 are now in the supplementary material, and are labelled as Fig S1, S2, and S3, respectively. Consequently, the numbers of the other figures have changed in the text.

REVIEWER # 2

An interesting report by Bedoya et al., in essence testing a hypothesis about potential role of spontaneous vibrational emissions by BMSB males in establishing autumn aggregations. Even though the results do not support the hypothesis, I believe such a negative result would be a valuable contribution to the field and an incentive for researchers to look for real mechanisms regulating aggregation behaviour of this important pest, as well as the real role of MS-1 signals. To a lesser extent, the report is also valuable as a partial replication of recordings by Polajnar et al. (2016), confirming similar vibratory behaviour of an American population in a different season, although this aspect requires improvement (see below).

Q6: Results lack more precise descriptions of recorded signals (L224-230); all temporal and frequency ranges should include basic descriptive statistics, terms like "occasionally", "usually" and "on rare occasions" must be avoided.

R6// We have now added details on the spectro-temporal description of the BMSB sounds, including descriptive statistics (Lines 232-239). We have also removed the ambiguous wording from the results.

Q7: More importantly, playback experiments are not reproducible from this report which lacks information about playback amplitude, at least at the source, but preferably at various points within the experimental arena - both in absolute terms and relative to signals emitted by live males. Amplitude could importantly affect attractiveness of artificial signals, either negatively or positively. Luckily, the issue is probably easy to resolve by simple calibrated measurement.

R7// Thank you for pointing out this clear omission from our manuscript. We have now used a laser vibrometer to measure playback amplitude at the source (both directly on the transducer and on the shelter) and at the middle of the screened cage (release point) (Lines 211-217). We currently do not have access to a BMSB population, so we could not measure displacement velocities of male signals using the laser vibrometer; fortunately, the reported amplitude values are enough to reproduce our experiments, which was the reviewer's main concern.

Q8: L205 needs precise information about the acoustic transducer used, and the report would benefit from including a sample stimulation sequence and a length of brown noise as supplementary sound files.

R8// We have now included a more detailed explanation of the transducer used (Line 207), and have also included a set of recordings of brown noise and BMSB vibrations (MS1 and MS1+MS2, raw and amplified) as supplementary files (note, these are low frequency vibrations and only audible with the proper equipment).

Additional minor comments:

Q9: L151-153: Which software for acoustic measurements?

R9// We performed all acoustic analyses using Matlab R2018b. We have now added this in the manuscript (Line 168).

Q10: L167: Please state exactly which "vibrations" were averaged. Consider including the average spectrogram as a figure.

R10// We averaged a representative subset of 15 BMSB vibrations. It consisted of MS1 signals mostly, and in some occasions of MS1+MS2, this based on what we found during our data collection. The average spectrogram of such signals was shown in figure 3, which is now figure S2, as requested by reviewer 1.

Q11: L181: Be careful about terminology - in a behavioural study, the word "signal" should be reserved for vibratory emissions used in communication to avoid confusion.

R11// The reviewer is quite correct! Sorry we missed this; we have exchanged the term signal for 'movement amplitude value' in order to avoid confusion.

Q12: L212-213: I assume "calls" mean MS1+MS2 (judging by duration longer than MS1 and by what is shown in Fig. 5), but please specify what exactly was played, preferably with a sound file included as a supplement.

R12// Good observation. We played a set of recordings with representative elements of what we found in the dataset, which was dominated by sporadic MS1 calls, sequences of MS1+MS1 calls, and sequences of MS1+MS2 calls. We have now added a set of recordings, including brown noise, to the supplementary material so that readers have a better idea of what vibrations were played back to the insects.

Q13: L256: "Ruling out" is perhaps too strong a statement, given that the authors didn't study this in detail.

R13// We have now changed this to "largely excluded".

Q14: L279-280: Any comment about possible association with aggregation pheromone production, which is also male-only and may indicate group size? A statement about whether you found any mating couples could also be informative.

R14// We never found any mating couples inside the shelters, and added this information in line 290. To date, there is no evidence that there is an overwintering pheromone in this species. The bugs tend to stop producing the aggregation pheromone as they disperse, and then stop responding to it over the winter (Morrison et al. 2017: Behavioral response of the brown marmorated stink bug (Hemiptera: Pentatomidae) to semiochemicals deployed inside and outside anthropogenic structures during the overwintering period. *J. Econ. Entomol.* **110**, 1002-1009. (doi:10.1093/jee/tox097). We have added some commentary about this in Lines 290-294.

Q15: L288-290: Discussion about sensory cues for cluster formation should mention the work of Toyama et al. (2006).

R15// We have now made a reference to Toyama et al.'s work in line 304.

Appendix B

Response to the Reviewers

We thank the reviewers for the time and effort they put into reviewing our manuscript. It has significantly improved due to their feedback. We have added the n values for the statistical descriptions and corrected the suggested sections.